# Selection and Phenotyping for Drought Tolerance in Somatic Hybrids between *Solanum tuberosum* and *Solanum bulbocastanum* That Show Resistance to Late Blight, by Using a Semi-Automated Plant Phenotyping Platform

**Tünde-Éva Dénes [1,\*], Imola Molnár [2], István Zoltán Vass [3], Imre Vass [3] and Elena Rákosy-Tican [2]**

[1] Biological Research Center Jibou, Babes-Bolyai University, 455200 Jibou, Romania
[2] Plant Genetic Engineering Group, Department of Molecular Biology and Biotechnology, Faculty of Biology and Geology, Babes-Bolyai University, 400006 Cluj-Napoca, Romania; molnar.imola5@gmail.com (I.M.); elena.rakosy@gmail.com (E.R.-T.)
[3] Institute of Plant Biology, Biological Research Centre, Hungarian Research Network, 6726 Szeged, Hungary; vassiz@gmail.com (I.Z.V.); vass.imre@brc.hu (I.V.)
\* Correspondence: denes.tundeeva@gmail.com

**Abstract:** Drought stress is one of the most limiting abiotic stresses for plant growth and development. Potato (*Solanum tuberosum* L.), due to its shallow root system, is considered sensitive to drought. In potato breeding, the wild *Solanum* species may represent a good resource for disease and abiotic stress resistance genes, but their transfer is limited by sexual incompatibilities. Somatic hybrids (SH) between potato and the wild species *Solanum bulbocastanum*, sexually incompatible with potato, proved to be late-blight-resistant in laboratory and field assays. The aim of this study was to screen a series of somatic hybrids and derived backcrosses for drought stress tolerance. In vitro stress exposure (with 5% and 15% PEG 6000) allowed the selection of several tolerant genotypes in a short time. The eleven selected genotypes were tested by using a semi-automated plant phenotyping platform at the Biological Research Centre in Szeged, Hungary, where the plants' biomass accumulation and photosynthesis under long-term drought conditions were monitored. The findings of this study affirm that the somatic hybrids between potato and *S. bulbocastanum*, along with their backcrosses, constitute valuable pre-breeding material. This is attributed to their possession of both late blight resistance and drought stress tolerance.

**Keywords:** potato somatic hybrids; drought; proline; photosynthesis; phenotyping; selection for stress-selection

## 1. Introduction

Global climate change increases the challenges in agriculture because abiotic stresses and related biotic stress factors induce pressures on crop yield. Potato (*Solanum tuberosum* L.) is one of the most essential cultivated plants of the world, the fourth in place after rice, wheat, and maize as a produced food crop [1–3]. Breeding programs, thus far, have focused on obtaining disease-resistant potato cultivars with higher yields. In the near future, climate change will cause a significant decline in the yields of potato production [4] The decline is already being felt due to the fact that the majority of the potato cultivars, including the modern ones, are sensitive to drought [5]. Therefore, it becomes extremely important to develop a drought-tolerant cultivar.

Potato is a drought-sensitive species capable of withstanding, in some cultivars, a moderate water deficit that can have a major impact on physiological processes, growth, and yield [6]. The shallow root system, with 85% of its roots being concentrated in the upper 0.3 m of soil, is the primary cause of drought sensitivity in potato crops [7,8]. The impact of drought stress depends on the timing, each growth stage of the potato crop

showing its characteristic response [9]. A decrease in leaf area and the number of leaves, a reduction of plant height, and a limitation of canopy expansion represent the primary visual changes during a time of water deficit. Early senescence due to water deficit results in premature flowering and reduced yield [9]. Even a short period of drought can cause reduced tuber production [10]. There is a correlation between the duration of water stress and the number of tubers per stem [11].

Wild *Solanum* species are considered reservoirs of resistance genes against diseases, pests, and abiotic stress. The screening of wild *Solanum* species yielded results indicating that *S. acaule*, *S. demissum*, *S. stenotonum*, and *S. gandarillasii* carry drought stress resistance genes, which can be exploited [12,13]. Additionally, *S. bulbocastanum* was identified as highly resistant to late blight caused by *Phytophthora infestans,* and four resistance genes were identified and cloned from different accessions of this species (AY303171, DQ122125, FJ188415, and FJ536346) [14–17].

Somatic hybrids (SHs) between the cultivated potato and the incongruent wild diploid species *S. bulbocastanum* were previously produced, and some of them proved to be fertile and backcrossed with potato [18]. Further studies demonstrated that many of the SHs and their progenies carry two durable resistance genes against late blight, *Rpi-blb1* and *Rpi-blb3*, along with two R genes *R3a* and *R3b*. These hybrids are resistant in detached leaf assay, maintain resistance in the field, and demonstrate favorable yield and processing qualities [19]. The effect of salt and drought stress on *S. bulbocastanum* physiological responses was investigated by Daneshmand [20], revealing that both stress factors reduced plant growth and photosynthetic pigment content and enhanced antioxidant enzyme activity.

The goal of combinatorial biotechnology is to combine multiple resistance traits with tolerances to abiotic stresses, such as drought, in order to address new challenges posed by climate change [21]. Generally, drought stress tolerance can be achieved either through in vitro stress selection experiments or through selection in a greenhouse or field by comparatively monitoring the phenotype in control or drought stress conditions. In in vitro conditions, culture media supplementation with osmotic agents such as sorbitol, mannitol, salt (NaCl), and polyethylene glycol (PEG) is commonly employed to simulate water stress [22]. The use of PEG with a molecular weight greater than 6000 (MW $\geq$ 6000) is recommended because it does not penetrate into plant cells, is non-ionic, and mimics dry soil conditions [22].

During drought stress, the proline level serves as a biochemical marker of the plant's adaption ability [23,24]. Elevated proline levels are a result of upward- and/or downward accumulation [25,26]. However, upon rehydration, the opposite effect can be observed. Proline also neutralizes free radicals, preventing oxidative stress and maintaining NAD(P)+/NAD(P)H ratios at a normal level [27].

In the greenhouse or field, the best way to monitor the phenotypic changes of plants under drought is by using phenotyping platforms or remote sensing equipment [28]. This approach allows for the collection of a large amount of useful data, which can be processed by using different software tools. Such processing facilitates the comparisons between control and stressed crop plants, highlighting important datasets [29]. Moreover, these new opportunities made it possible to combine phenotyping with genotyping data and create a knowledge-based selection of the best genotypes for breeding [30].

The aim of our research was to identify somatic hybrids between potato and *S. bulbocastanum* that possess both late blight resistance and drought stress tolerance. In the first stage, in vitro selection with increasing concentrations of PEG 6000 was applied at the plantlet level to seven somatic hybrids and eight BC$_1$- and three BC$_2$-derived clones, in comparison to their parents. Subsequently, the selected plants were transferred to the HAS-RSDS Szeged EPPN phenotyping platform, where plant biomass (green canopy) and photosynthetic performance were recorded under both control and drought-stressed conditions. To the best of our knowledge, this article represents the first report on the phenotyping of somatic hybrids of potato + *S. bulbocastanum* under drought stress.

## 2. Materials and Methods

The somatic hybrids between the cultivated potato *Solanum tuberosum* cv. Delikat and *Solanum bulbocastanum* GLKS-31741 (blb41; Gross Lüsewitz Potato Collections (GLKS)), of the IPK Gene bank in Germany, were obtained by electrofusion of the protoplast [18]. The Delikat cultivar is characterized by early maturation of oval tubers, superficial eyes, yellow skin, and a pale yellow flesh (Thieme et al., 2008) [31]. The *S. bulbocastanum* clone was identified as carrying specific resistance genes, Rpi-blb1, Rpi-blb3, R3a and R3b.

The parental species were cultured in vitro on RMB5 medium [32] (MS salts and B5 vitamins, Duchefa Biochemie, Harleem, The Netherlands), using test tubes. Protoplast isolation, electrofusion, and culture were conducted following established protocols, with modifications in enzyme concentrations for different potato cultivars. Electrofusion conditions and culture parameters were presented in detail by Rákosy et al. (2015) [18]. Only hexaploid putative shoots exhibiting vigorous growth were selected for further cultivation and flow cytometry analysis, while mixoploid or tetraploid shoots were discarded during this stage of selection.

Plants were grown at 21 °C, with a 16 h photoperiod and 135 μmol m$^{-2}$ s$^{-1}$ light intensity in a climate chamber. The initial clones and their characteristics were as determined in previous experiments [18,19], where are shown the presence of resistance genes, results after agroinfiltration, and field test results; detailed results are presented in Supplementary Table S1. Agroinfiltration was performed using a culture of the *Agrobacterium tumefaciens* strain AGL1 + pVirg transformed with a pK7WG2 plasmid containing the avirulence factors for Rpi-blb1 (Avrblb1), Rpi-blb3 (Avr2—recognized also by R2), R3a, and R3b (Avr3a and Avr3b), obtained from the University of Wageningen [18] (Supplementary Table S1).

### 2.1. In Vitro Drought Stress Selection

To simulate drought stress, the culture media were supplemented with polyethylene glycol (PEG 6000 AppliChem GmbH, Darmstadt, Germany) [33], which changes the osmotic potential and causes a water deficit to the plant. The drought stress in vitro simulation was performed in two steps: first, the culture medium was supplemented with 5% PEG 6000-mild stress and in a second step with 15% PEG 6000-severe stress. The duration of the treatments was three weeks. In each experiment, two groups of plants were used: the control group cultured on RMB5 (MS salts with vitamin B5 medium, Duchefa) without PEG and the treated group (*n* = 5).

At the end of each experiment, the morphological traits of the plants were measured: the number and length of roots and stems as well as the proline concentration. The proline concentration was quantified from plant leaves [34], using the protocol developed by Bates [35]. The plant material (60–80 mg) was ground with 3% salicylic acid (5 μL/g plant). After centrifugation at maximum speed, for 10 min, 100 μL of the supernatant was transferred into a new tube which contained 100 μL sulphosalicyclic acid, 200 μL acid-ninhdrin, and 200 μL glacial acetic acid. The mixture was incubated for 1 h, at 95 °C, being subsequently quickly cooled on ice. To each sample, 1 mL toluene was added, and the mixture was stirred vigorously. The mixture separated into two phases. The optical density of the upper phase was measured at 520 nm on a JENWAY 6105 UV/VIS spectrophotometer, with toluene used as a blank.

The proline quantity was calculated after the following formula:

$$\mu\text{moles/g tissue} = [(\mu\text{g proline/mL} \times \text{mL toluene})/115.5 \ \mu\text{g/}\mu\text{mole}]/[(\text{g sample})/5]. \tag{1}$$

### 2.2. Statistical Analysis

The results were statistically analyzed using Microsoft Office Excel 15, an R statistic program, version 3.3.1 car package. Measurement results of the shoot and root length and proline concentration were compared with Student's *t* test. Data from all measurements were normally distributed and had equal variance. One-way ANOVA was used to analyze biomass accumulation. We consider significant difference at the $p < 0.05$ level.

### 2.3. Biomass Accumulation of Somatic Hybrids under Drought Stress Ex Vitro

All measurements and investigations were performed by using the Phenotyping Platform HAS-RSDS at the Biological Research Centre, Szeged, Hungary, described by Cseri et al. (2013) and Spika et al. (2022) [36,37]. In vitro grown plants were transferred in pots containing 50% sandy (Maros) soil and 50% Terra peat soil. An equal quantity of chemical fertilizer (SUBSTRLAL®, Osmocote Plus®, Wals-Siezenheim, Austria) was added to each pot. In the drought stress experiment, three pots for each somatic hybrid were exposed to water deficit, and three others were treated as control. After two weeks, when plants were acclimatized and covers (plastic beakers) removed, the plants were thinned, one being left in each pot. The soil water capacity was determined, and pots were watered to 20% (water limitation) and 60% (control, well-watered) of the 100% soil water capacity. Watering was done automatically by a plant moving system including balance, in connection with a computer-mediated peristaltic pump. Each pot was equipped with a radio-frequency identifier (RI). The growth of plants was monitored by digital photography, each plant being photographed by an Olympus C-7070WZ (Olympus Ltd., Southend-on-Sea, UK) digital camera from eleven different sideway positions, produced by 32–33 step rotations of the pot.

Plant biomass was determined based on the images, subtracting the homogenous background and counting the green pixels. Monitoring the plant growth was performed for six weeks. After the experiment, the total weight of tubers was measured in drought-stressed and in control plants. Tubers were harvested 75 days after planting (the completion of the growth period); the mean total weight was measured, and the form of the tubers was evaluated. Data analysis was performed with the Matlab software tools with the Image Processing Toolbox Ver. r2013b (The MathWorks Inc., Natick, MA, USA).

### 2.4. Evaluation of Drought Stress Effect on Photosynthesis

Chlorophyll fluorescence emission was measured with a pulse amplitude modulation fluorometer (PAM-2000 Heinz Walz Gmbh, Effeltrich, *Germany*) and pocket Plant Efficiency Analyzer (PEA) chlorophyll fluorimeter (Hansatech Instruments, King's Lynn, UK). The measurements were performed twice, the first time in the second week and then in the fifth week of the experiment, in order to follow the changes of the parameter of interest. Before use of the pocket PEA, plants were dark-adapted for at least 15 min, for precise determination of the minimal–maximal fluorescence levels in the dark (Fo, Fo' and Fm, Fm') and the performance index [38]. Mini-Pam instruments were used to measure photo-synthetically active radiation (PAR), electron transport rate (ETR), maximal fluorescence yield, and effective quantum yield.

From the chlorophyll a fluorescence transients, the following data were acquired: the (Fo) initial fluorescence level (measured at 50 μs) and P (Fm) maximal fluorescence intensity, as well as the J (at about 2 ms) and the I (at about 30 ms) intermediate fluorescence levels. From these specific fluorescence features, the following parameters of photosynthetic efficiency were calculated: maximal PSII quantum yield, Fv/Fm; the ratio of variable fluorescence to initial fluorescence, Fv/Fo, where Fv = Fm − Fo; Vj = (F2 ms − Fo)/Fv; and the performance index $PI_{Abs}$ [39,40].

$$PI_{Abs} = 1 - (Fo/Fm)/(Mo/Vj) \times (Fm-Fo)/Fo \times (1 - Vj)/Vj, \qquad (2)$$

where Mo = 4 × (F300 μs − Fo)/(FM − Fo) represents the initial slope of fluorescence kinetics.

The performance index calculated at the beginning (2nd week) and at the end of the experiment (5th week) were used to obtain the Drought Factor index (DFI) [41].

$$DFI = \log A + 2 \log B, \qquad (3)$$

where A is the relative PI measured in the beginning of the experiment, and B is the relative PI measured at the end of the experiment.

The relative PI was determined using the following formula: $PI_{drought}/PI_{control}$.

## 3. Results

### 3.1. In Vitro Drought Stress Selection

Based on morphology, specifically, of stem and root development, genotypes were classified into four different groups. The first group consists of plants with well-developed stems and roots (group 1). The second group comprises members with weakly developed stems, no presence of mini-tubers, and weak root development (group 2, 15% reduction in length compared to group 1). The third group contains genotypes with developed roots but without stem development (group 3). Finally, the fourth category includes genotypes that did not exhibit any stem or root development, and some of them died during the experiment (Table 1, classification of all investigated genotypes).

**Table 1.** Somatic hybrids and parental species classification based on morphologic traits after in vitro drought stress application (MS + 5%PEG and MS + 15% PEG).

| Treatment | Group 1 Genotypes | Group 2 Genotypes | Group 3 Genotypes | Group 4 Genotypes |
|---|---|---|---|---|
| Media supplemented with 5% PEG | SH 99/2<br><br>SH 84/5<br>SH 95/1<br>$BC_1$ 83/9/64<br>SH 83/9<br>SH 82/4<br>$BC_2$ 82/4/68/22<br>$BC_1$ 83/9/63<br>$BC_1$ 83/9/3 | SH 82/4/68<br><br>SH 94/5<br>$BC_2$ 95/1/4/11<br>$BC_1$ 82/4/4<br>$BC_1$ 82/4/38 | *Solanum tuberosum* cv. Delikat<br>*Solanum bulbocastanum*<br>$BC_1$ 94/5/5 | - |
| Media supplemented with 15% PEG | - | SH 94/5<br><br>$BC_1$ 94/5/5<br>$BC_2$ 95/1/4/11<br>$BC_2$ 82/4/68/22<br>$BC_1$ 95/1/3<br>SH 99/2<br>SH 84/5<br>SH 82/4 | SH 83/9<br><br>$BC_1$ 83/9/63<br>*Solanum tuberosum*<br>*Solanum bulbocastanum*<br>SH 95/1<br>$BC_1$ 95/1/7 | $BC_1$ 83/9/27<br><br>$BC_1$ 83/9/64<br>$BC_2$ 95/1/4/59 |

The shoot lengths in group 1 are significantly greater (*t* test, $p < 0.05$) than the shoot lengths in group 3; similarly, the group 1 root lengths are significantly greater than the root lengths in group 2 (test, $p = 0.0013$) (mean shoot length group 1 = 2.63 cm; mean shoot length group 3 = 2.13; mean root length group 1 = 1.35; mean root length group 2 = 1.1). These results prove that the genotypes from group 1 are more tolerant of drought stress than genotypes from other groups.

Potato is a drought-sensitive plant, and our results support this fact by revealing that mild drought stress has a negative impact on the development of somatic hybrids. As mentioned above, the development of drought-stressed plants shows signs of stress when compared to that of the control plants. Classification of genotypes based on morphological traits helped us to understand the genotypes' resource allocation.

In cases of severe drought stress (MS + 15% PEG), plants showed poor development, while the control plants' development was normal. Using the same classification criteria as in mild stress, in this case, no genotypes can be framed in group 1 (Table 2). There are a few genotypes in group 2 that show 1–2 cm growth of the stem or abaxial stem, while members of group 3 have a slightly developed root system. Classification of plants based on their morphological traits is in accordance with the scientific literature. Stem and root development under drought stress was assessed in many research papers, because this trait adequately reflects the effects of abiotic stress on plants [42].

**Table 2.** Drought stress effects on tuber production in somatic hybrids (potato + *S. bulbocastanum*) and parental lines (up arrow—increase; down arrow—reduction), with percentage compared to the unstressed control.

| Genotype | Generation | Yield Changes % |
|---|---|---|
| *Solanum tuberosum* cv. Delikat | parent | 53 ↓ |
| *Solanum bulbocastanum* | parent | No tubers |
| SH 83/9 | Hybrid | 83.63 ↓ |
| BC$_1$ 83/9/3 | BC1 | 72.81 ↓ |
| SH 99/2 | Hybrid | 96.75 ↓ |
| SH 82/4 | SH | 85.27 ↓ |
| BC$_1$ 82/4/68 | BC1 | 21.38 ↓ |
| BC$_2$ 82/4/68/22 | BC2 | 100 ↓ |
| SH 95/1 | Hybrid | 90.72 ↓ |
| BC$_1$ 95/1/7 | BC1 | 48.58 ↑ |
| BC$_2$ 95/1/4/11 | BC2 | 71.66 ↓ |

### 3.2. Proline Content

The proline content of the drought-stressed group is significantly increased for both PEG concentrations applied (*t* test, $p < 0.01$), but the quantity of proline in mild and severe drought stress is genotype-dependent. The difference between the parental species' response to drought stress is remarkable. In the wild species, proline gradually increases according to the drought severity, while in the cultivated potato, the elevated proline content is present only in the mildly drought-stressed plants, and in severe drought stress conditions, the plants produce only a small amount of proline (Figure 1).

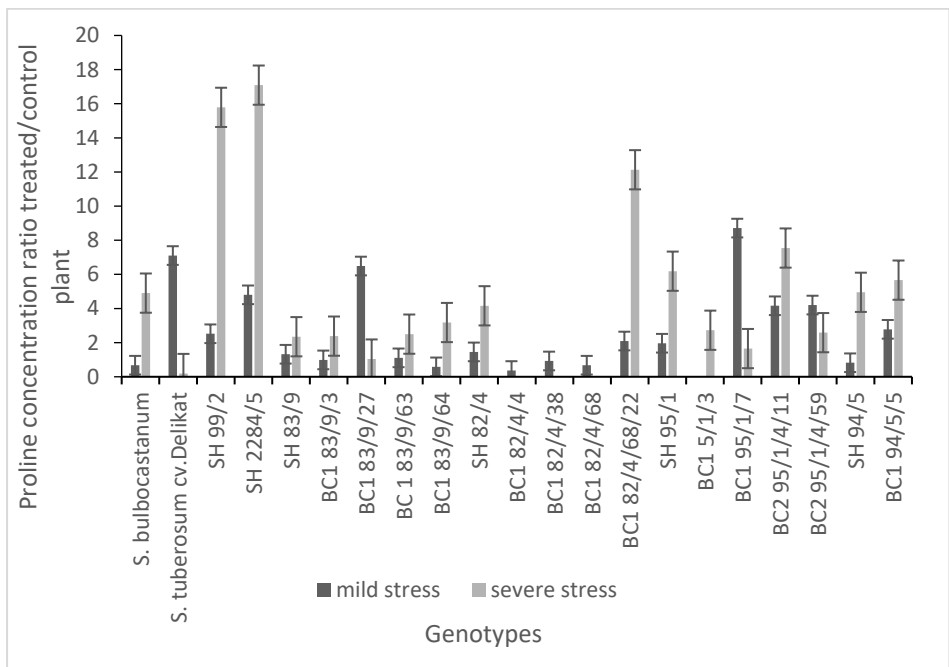

**Figure 1.** Proline ratio in response to mild or severe drought stress simulated in vitro with PEG 6000 (5% mild; 15% severe stress).

### 3.3. Ex Vitro Plant Development in Somatic Hybrids under Drought Stress

Drought stress significantly decreased the biomass accumulation of the cultivated potato cv. Delikat (Figure 2A) (ANOVA, $p = 0.003$), while *S. bulbocastanum* (Figure 2B) and somatic hybrids (Figure 2C,D) show no significant differences between control and drought-stressed plant development (ANOVA, $p = 0.059$). In the investigated genotypes, drought-stressed plants showed slower canopy expansion (Figures 2 and 3).

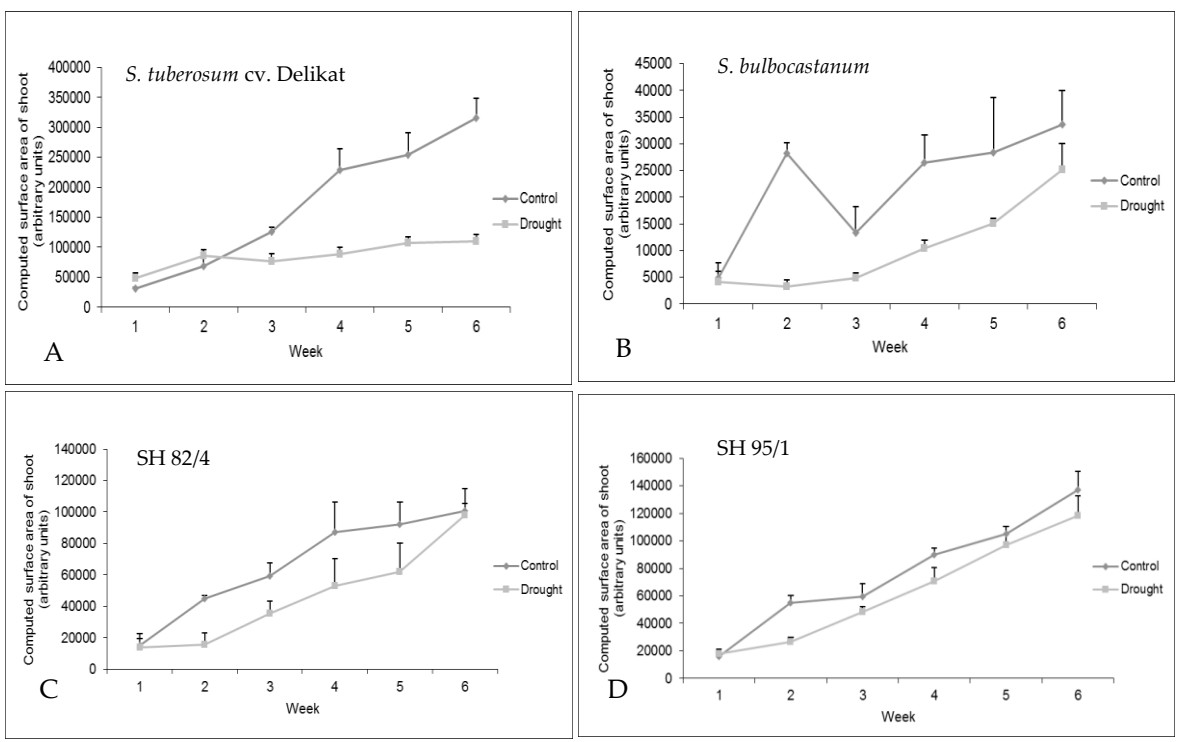

**Figure 2.** Biomass accumulation in *Solanum tuberosum* cv. Delikat (**A**), *Solanum bulbocastanum* (**B**), SH 82/4 (**C**), and SH 95/1 (**D**) genotypes (Mean + standard deviation (SD)).

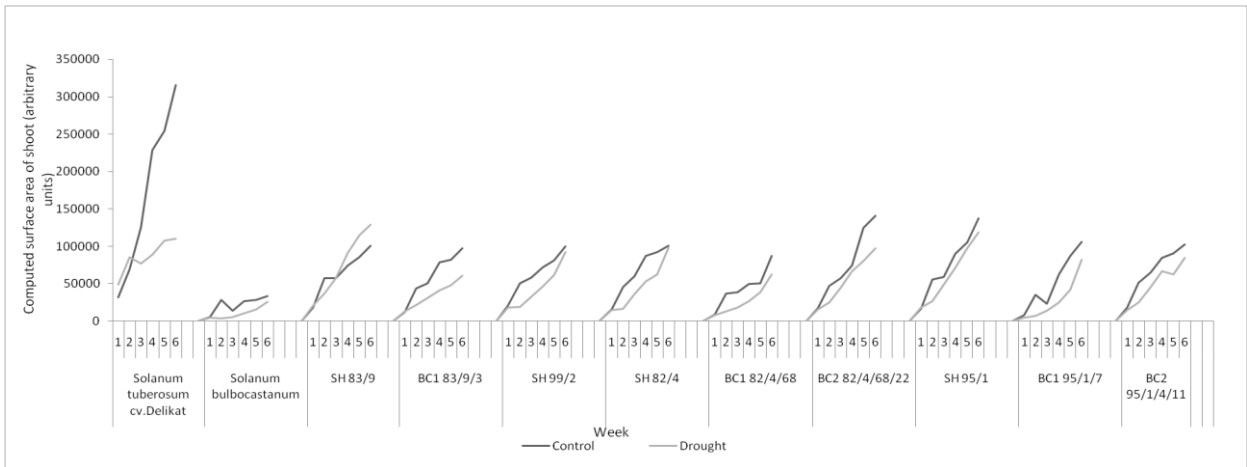

**Figure 3.** Biomass accumulation in parental lines and somatic hybrids during the six weeks of the experiment.

### 3.4. Tuber Yield

Tuber yield measurements in somatic hybrids, backcross progenies, and parental lines revealed that drought stress has significantly reduced tuber yield as compared to the control group (*t*-test, $p = 0.0051$). Although the yield is reduced, in the case of certain genotypes, the average tuber size is preserved, for example, in $BC_1$ 82/4/68 and $BC_1$ 95/1/7 (Figure 3). This is due to the different genetic background of the somatic hybrids [19]. Despite the drought stress, only a small extent of yield reduction was observed in the following genotypes: SHs 99/2 and 83/9 and $BC_1$ 83/9/3 (Figure 4).

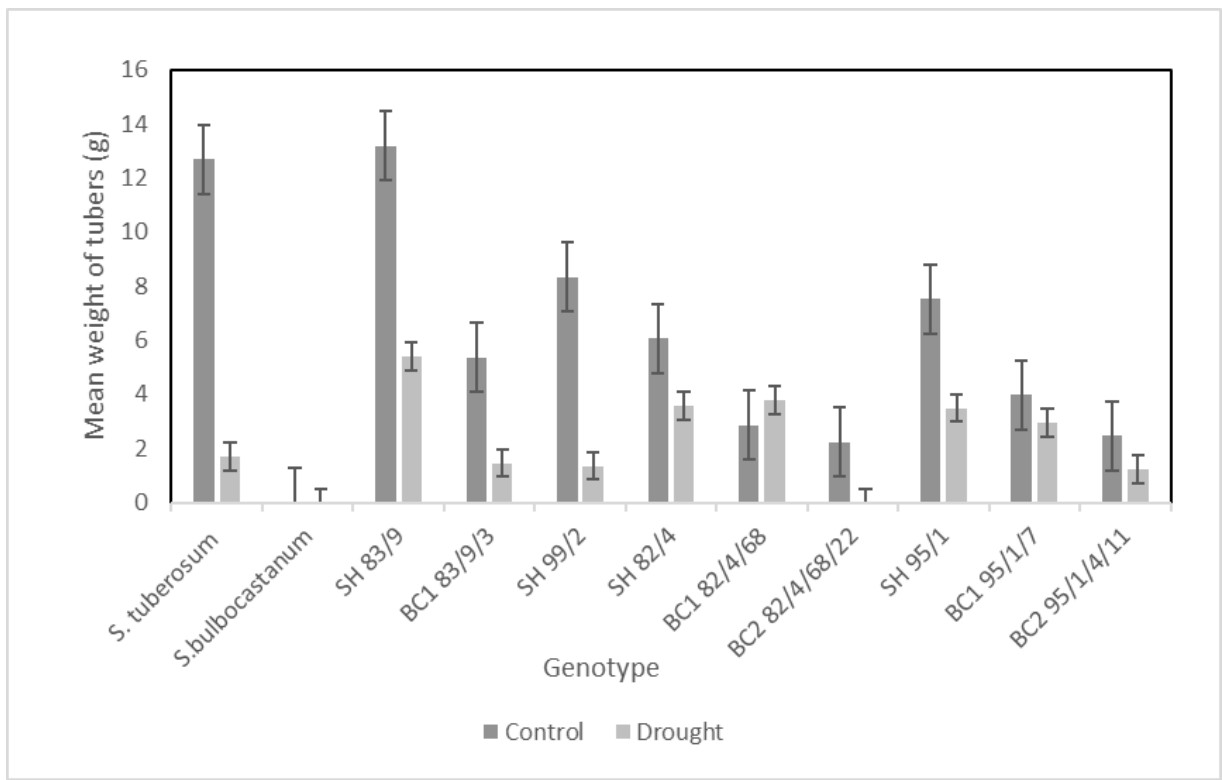

**Figure 4.** Mean weight of one tuber of different genotypes (3 plants/genotype) after six weeks of water shortage.

One goal for the future is to conduct field tests to gather information on the shape, quantity, and quality of tubers under different drought stress conditions. The importance of potato size and total yield depends on specific objectives and the utilization area. In terms of the commercial aspects of potatoes, size generally plays a more decisive role. Larger potato sizes are typically advantageous as they are more appealing to consumers and better align with consumer preferences. The significance of total yield can also be substantial, since growers usually aim to harvest the highest possible quantity of potatoes within the available area. Total yield influences profitability and economic viability, as well as production goals and market demands.

As already mentioned, from the point of view of agriculture, it is possible to make two types of ranking based on total production under drought stress. The $BC_1$ 95/1/7 genotype produces more tubers than the control, and in the $BC_1$ 82/4/68 genotype, production was only reduced by 21% (Table 2). If the size of the tubers is important, the BC1 82/4/68 genotype is ranked first because under drought stress, the tuber size increased. The $BC_1$ 95/1/7 genotype would be in second place.

*3.5. Effect of Drought Stress on Photosynthesis*

Quantification of the maximal quantum efficiency (Fv/Fm) revealed that at the time of the first measurements (2nd week of the experiment), a slightly increased ratio for the stressed plants can be observed in all genotypes.

Significantly greater values were observed only in four genotypes: *S. bulbocastanum*, SH: 99/2 and 83/9, and $BC_1$: 83/9/3 (*t* test, $p = 0.003$; $p = 0.003$; $p = 0.003$; $p = 0.03$). During the second measurements (5th week of the experiment), no significant difference could be detected between the genotypes (Figure 5).

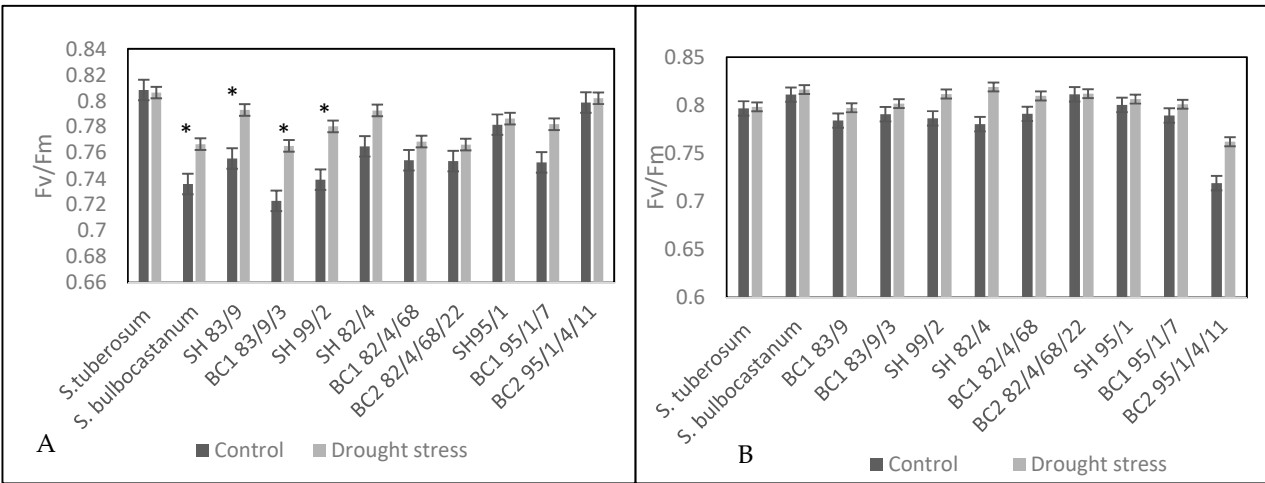

**Figure 5.** First measurement in the 2nd week (**A**) and second measurement in the 5th week (**B**), the results of maximal quantum efficiency in control and treated genotypes; the significant difference ($p < 0.05$) between control and stressed groups were marked with a star.

The effective quantum yield (Fv'/Fm') was determined by using the light-adapted values of initial and maximal fluorescence. The effective quantum yield declines significantly in the drought-stressed group at the first measurement (*t* test, $p < 0.01$). On the other hand, at the second measurement, no differences between the control and the stressed group were observed (Figure 6).

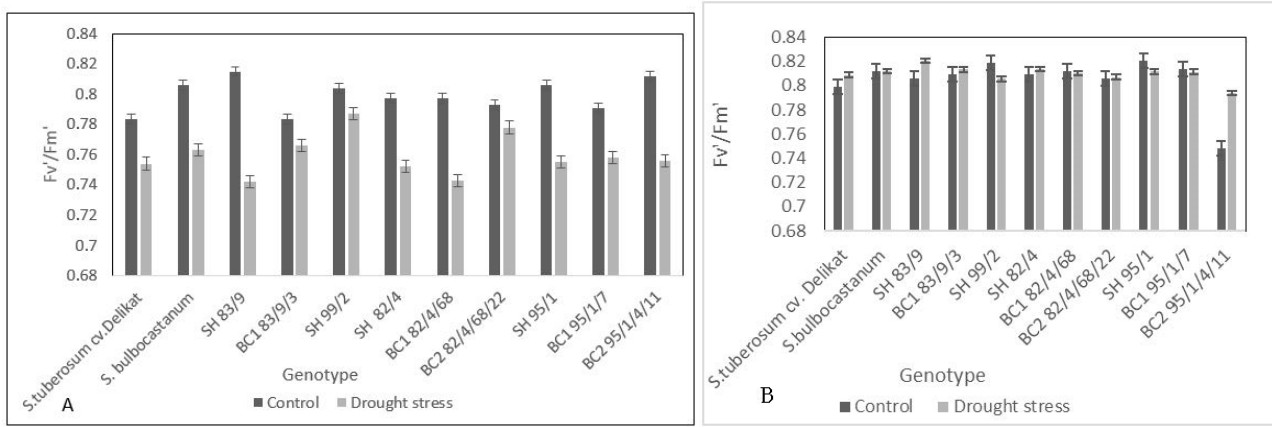

**Figure 6.** First (**A**) and second (**B**) measurement results of effective quantum yield in control and treated genotypes.

Another important parameter was the Fv/Fo, which reflects the efficiency of the water-splitting complex, which is also a sensitive component in the photosynthesis process, according to recent findings [43]. This fact is demonstrated by our results from the first measurement (2nd week), the control plants being characterized by an increased value compared to the treated group (Figure 7). Significant differences were found in four genotypes: *S. bulbocastanum*, SH 83/9, BC$_1$ 83/9/3, and SH 99/2, similar to the maximum quantum efficiency results (*t* test, $p < 0.05$).

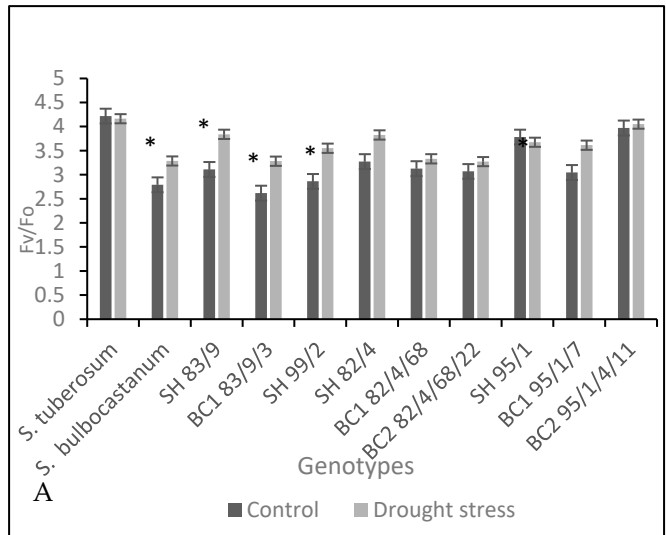
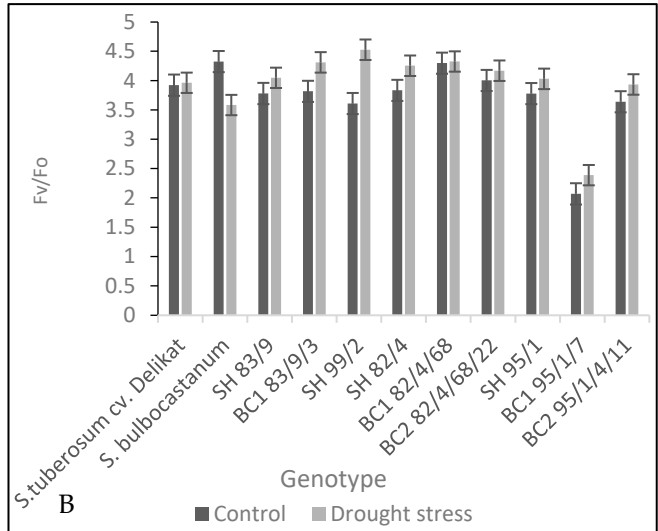

**Figure 7.** First measurement result of efficiency of water-splitting complex (Fv/Fo); first (**A**) and second (**B**) measurement results; the significant differences ($p < 0.05$) between control and stressed groups were marked with a star.

The performance index, an integrative parameter, reflects the activity of PSII, and therefore it can be used to investigate all of the photosynthesis process mechanisms. First-time measurements (2nd week) of the performance index showed no differences between the control and the drought-stressed plants (Figure 8A). In the second measurement, we observed an increase of the performance index in most of the drought-stressed lines, with the exception of *S. bulbocastanum* and SH: 99/2 (*t* test, $p > 0.01$) (Figure 8B).

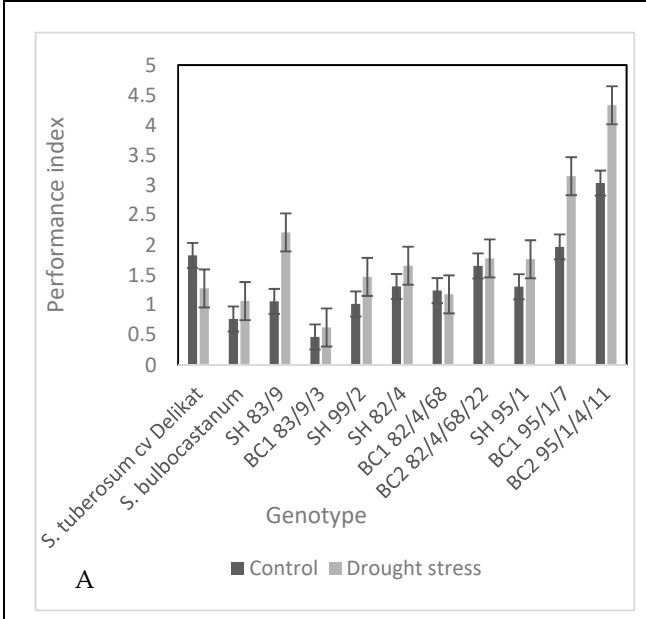
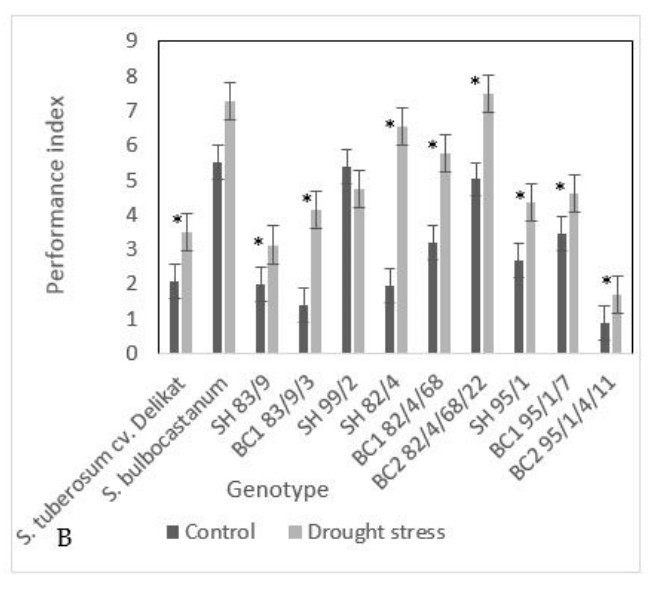

**Figure 8.** Performance index of somatic hybrids and parents in control (black columns) and drought-stressed (grey columns) plants, first (**A**) and second (**B**) measurement results the significant differences ($p < 0.05$) between control and stressed groups were marked with a star.

The decrease of Fv/Fo ratio in drought-stressed plants indicated the perturbation of the water-splitting complex activity. In the case of the second measurement (5th week), no significant difference between the control and the treated groups was found (*t* test, $p > 0.05$),

these findings suggesting that our somatic hybrids have the capacity to adapt to water scarcity, Figure 7.

## 4. Discussion

The selection of potato genotypes with resistance to late blight and tolerance to drought stress represents an important target in potato breeding. Historically, drought tolerance was not a primary objective in breeding programs, which mainly concentrated on developing disease-resistant cultivars [2]. The lack of information on the drought tolerance of wild *Solanum* species has hindered the identification of promising candidates for breeding. An important study by Coleman [13] evaluated the drought tolerance of *S. gandarillasii*, a wild species known for its water-saving strategies, resulting in reduced biomass compared to the well-known *S. tuberosum* Kennebec cultivar under water deficit conditions.

Somatic hybrids between cultivated potato and *S. bulbocastanum* have proven to be resistant to late blight in detached leaf assay (DLA) and field tests [19], and many of them carry resistance genes known to confer durable resistance to late blight, i.e., *Rpi-blb1* and *Rpi-blb3*, along with two R genes *R3a* and *R3b* (Supplementary Table S1). In our current work, it was demonstrated that these somatic hybrids and some of their backcrosses exhibit tolerance not only to late blight infection but also to drought stress in greenhouse conditions. *S. bulbocastanum* is a tuber-bearing species, but during our experiment in greenhouse conditions, it did not developed tubers. The chosen method, an in vitro preselection for drought tolerance, using culture media with PEG, proved to be a time-saving method, which allows the screening of a wide number of genotypes in a short time.

Plant response to water deficit is a well-studied phenomenon. Deblonde and Ledent (2001) [10] found that drought significantly reduces stem height and the number and length of green leaves. These effects of drought stress were also observed in potato + *blb*41 SHs. Based on the results after mild- and severe drought stress selection in vitro, eleven genotypes were selected for assessment of their drought tolerance in greenhouse conditions using a phenotyping platform.

Phenotyping platforms offer a valuable opportunity to evaluate morphological and physiological changes in plants over time and under drought stress [2,37,44]. The presence of drought tolerance in some somatic hybrids supports the observation that these plants rapidly adapt to new environmental conditions. At the end of the experiment, some of them, for example, BC$_1$ 82/4/68 and BC$_1$ 95/1/7, demonstrated the ability to catch up with the control plants, not only in green biomass but also in the mean size of tubers or productivity, which is essential to breeders. These backcross progenies also carry genes Rpi-blb1, Rpi-blb3, R3a and R3b which confer resistance to late blight [18,19].

By assessing both green biomass and tuber yield, we obtained a comprehensive indicator of drought stress. In accordance with other data, the main effects of drought stress observed in the development of our SH lines were the reduced canopy diameter and the reduced leaf and tuber numbers [9,45]. Biomass determination based on image technology proved reliable, as demonstrated by the high correlation between the plant's actual biomass and the biomass predicted on the basis of image-derived green pixels [44].

It is noteworthy that some somatic hybrids respond better to drought stress than the parental lines. Somatic hybrids represent a good example of a polyploid plant, in the sense that it is characterized by heterosis, a phenomenon that occurs due to the increased ploidy and the forced coexistence of different nuclei in one cell, which lead to gene expression alteration [46]. Thus, in addition to the transference of tolerance from the wild species *blb*41, heterosis is another probable cause of higher tolerance to drought stress in our somatic hybrids when compared to the cultivated cultivar. Various genotypes demonstrate the ability to produce more tubers even under drought stress than the cultivated potato, for example, the SHs: 82/4, 95/1, and 83/9 and BC$_1$: 95/1/7. Notably, BC$_1$: 82/4/68, even under drought stress, demonstrates the ability to develop more tubers than the non-stressed plants. Tolerant genotypes exhibited temporally slower biomass growth in the second week

of the experiment when the cultivated potato manifest the first symptoms of sensitivity. There are also differences between the green biomass growth rate of the SHs and BCs.

Fluorescence measurements indicated that the maximum quantum yield and effective quantum yield cannot serve as indicators of drought stress in potato and its SHs and BCs with *S. bulbocastanum*, as similarly demonstrated in barley [47]. As Flagella et al. (1998) [48] mentioned, mild and moderate drought stress did not significantly decrease the quantum yield of PSII, and so the Calvin cycle remains just slightly affected [49,50]. Research by Lu and Zhang [51] revealed that drought does not affect the photochemical quenching, but it increases the non-photochemical quenching, playing a crucial role in protecting photosynthesis reaction centers. The stability of the maximal quantum efficiency was also demonstrated by Lu and Zhang [52] in wheat (*Triticum aestium* L.) under moderately and severely drought-stressed plants. Several articles reveal that for different species, different values or intervals of maximum quantum efficiency are obtained, which are characteristic for healthy, unstressed plants [52–54].

One useful calculated parameter is the performance index (PI), combining the three main functional steps taking place in PSII (light energy absorption, excitation energy trapping, and conversion of excitation energy to electron transport) and serving as a measure of drought stress tolerance [55]. PI, which provides useful and quantitative information about the physiological conditions and the vitality of plants, reveals differences among varieties under drought stress [41]. The performance index also offers data on the density of the active reaction centers, the probability of the electron being trapped in the photosynthetic reaction center, and the efficiency of the electron transport [39]. Numerous studies reported that these parameters are more sensitive to water stress than the maximal quantum efficiency [49,56]. Ghobadi [56] compared the Fv/Fm and PI changes due to drought stress in the sunflower (*Heliantus annus*), finding PI to be the more sensitive parameter. This was also true in our experiments, as opposed to Fv/Fm (and Fv'/Fm'), which showed a minimal difference between the well-watered and drought-stressed plants after the 5-week adaptation to drought conditions (Figures 4 and 5). PI values demonstrated a weak negative correlation with both the loss of leaf biomass obtained from digital phenotyping and the mean tuber weight.

## 5. Conclusions

In conclusion, global climate change poses a critical threat to agriculture, particularly affecting crucial crops like potatoes through induced abiotic and biotic stresses. As the fourth most vital food crop globally, potatoes face vulnerability to climate-related challenges. Wild *Solanum* species, which carry resistance genes, offer an opportunity for developing drought-tolerant cultivars. Somatic hybrids, between cultivated potatoes and *S. bulbocastanum*, show promise in combining late blight resistance and drought stress tolerance. Combinatorial biotechnology, with a goal to integrate multiple resistance traits, provides a potential solution to climate change challenges. Utilizing PEG in vitro, beside the study of physiological responses, offers insights into drought stress adaptation. Phenotyping platforms effectively monitor plant responses to drought stress in various conditions. This research focuses on identifying somatic hybrids with late blight resistance and drought stress tolerance, a crucial step in breeding climate-resilient potatoes with enhanced yield stability.

**Supplementary Materials:** The following are available online at https://www.mdpi.com/article/10.3390/agriculture14010048/s1, Table S1: Somatic hybrids and backcross progenies selected for the assay of drought tolerance and their resistance to late blight in detached leaf assay (DLA), in a field (Δ rAUDPC), and results of resistance gene presence by using specific molecular markers.

**Author Contributions:** E.R.-T. was involved in planning and supervised the project. T.-É.D. carried out the in vitro experiments and ex vitro experiments. I.V. was involved in the planning, and supervised the phenotyping experiments. I.M., T.-É.D. and I.Z.V. performed the photosynthesis measurements T.-É.D. performed the calculations and wrote the manuscript. All authors discussed

the results and contributed to the final format of manuscript. All authors have read and agreed to the published version of the manuscript.

**Funding:** T-É.D. was supported by funds from POSDRU/159/1.5/S/133391. I.M. and E.R.-T. are grateful to EPPN (European Plant Phenotyping Network, Grant Agreement No. 284443) for access to the phenotyping platform HAS-RSDS-SSDS Biological Research Centre Szeged, Hungary).

**Institutional Review Board Statement:** Not applicable.

**Data Availability Statement:** Data are available from the authors.

**Conflicts of Interest:** The authors declare no conflict of interest.

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
