# Peer review of "Selection and Phenotyping for Drought Tolerance in Somatic Hybrids between Solanum tuberosum and Solanum bulbocastanum That Show Resistance to Late Blight, by Using a Semi-Automated Plant Phenotyping Platform"

_agriculture, doi:10.3390/agriculture14010048_

Round 1

Reviewer 1 Report

Comments and Suggestions for Authors

Reviewer 2 Report

Comments and Suggestions for Authors

The manuscript required proofreading to avoid typographical errors, which I have enlisted
below for the corrections. Furthermore, the work presented is not enough to prove that these somatic hybrids from
S. tuberosum and S. bulbocastanum tolerate drought stress, so I suggested a major revision.
Minor Comments:
1. Line no. 93-95, please correct this sentence. upregulation and downregulation of proline
biosynthesis. Bteer to replace with up and/or down accumulation.
2. Lin no. 153-154, there are two citations, but the authors mentioned only one (Cesari et al); either
remove one citation or mention both citations
3. Line no. 157 ‘three pots for each genotype’ three plants from the same genotype have been
used, please correct this sentence as by removing ‘genotype’ word with ‘somatic hybrid
plant’ or ‘plant’
4. Line no. 250-251 (figure 2), please mention genotype names in Figure 2 A, B, C, and D as
well. Figure 2A, please correct the borders are not uniform with Fig 2 B, C, D and A
denotation is not properly placed in the A figure.
5. Figure 4 the ‘genotype’ word is not visible, and some letters mixed with ‘BC182’. Please
put it appropriately in the figure and correct ‘S. tuberosum’ it is mixed with the mean weight
of tubers
6. Line no. 281 in the results section, ‘effect of drought stress on the photosynthesis’ it should be
subsection 3.5; please correct it
7. Figure 6 A and B, the figure is denoted by A and B at two places. Please remove one from the
graph, and in Figure B, S. bulbocastanum is changed with S.tubersosum and vice-versa in
Fig A, please make corrections in this figure
8. Line no. 312-313, where is the figure for the second measurement result of the efficiency of
water splitting complex (Fv/Fo), please add the figure and place it with fig 7 and give
denotation as fig 7A and B
9. Line no. 323, please correct it should be ‘figure 8A’

10. Figure 8A, in every figure, you have written S. bulbocastanum, but here it is Solanum
bulbocastanum, keep ‘S. bulbocastanum’ consistent in every figure and ‘S. tubersoum’ is
written in other figures, but here it is S. tubersoum CV Delikat; please correct this, and
keep S. tuberosum consistent in every figure
11. Rakosy-Tycan et al (2020) showed these somatic hybrids from S. tuberosum and S.
bulbocastanum is resistant to late blight infection, and in your study, it was proved that
these somatic hybrids demonstrate tolerance to drought stress, and you have not done any
studies to prove late blight infection resistance in these somatic hybrids, so I suggest to
make changes in the title, ‘remove resistant to late blight infection’ or modify in such a
the way that it should only reflect your results and study about drought stress tolerance
Major Comments: In this research, authors have concluded that these somatic hybrids are tolerant to drought
stress based on only a few parameters (largely physiological parameters), such as proline content,
biomass accumulation and drought stress effect on photosynthesis.  1. I think these parameters are not enough to prove drought-tolerant plants, so I would suggest the
authors to do check few more physiological parameters, such as the relative water content in
leaf, leaf electrolyte leakage, stomatal density, Antioxidant enzymatic activities in leaf
extracts, Malondialdehyde (MDA) content, etc. 2. Last but not least, if possible, some transcript studies related to drought stress.

Comments on the Quality of English Language

The manuscript required proofreading to avoid typographical errors, which I have enlisted
below for the corrections. Furthermore, the work presented is not enough to prove that these somatic hybrids from
S. tuberosum and S. bulbocastanum tolerate drought stress, so I suggested a major revision.
Minor Comments:
1. Line no. 93-95, please correct this sentence. upregulation and downregulation of proline
biosynthesis. Bteer to replace with up and/or down accumulation.
2. Lin no. 153-154, there are two citations, but the authors mentioned only one (Cesari et al); either
remove one citation or mention both citations
3. Line no. 157 ‘three pots for each genotype’ three plants from the same genotype have been
used, please correct this sentence as by removing ‘genotype’ word with ‘somatic hybrid
plant’ or ‘plant’
4. Line no. 250-251 (figure 2), please mention genotype names in Figure 2 A, B, C, and D as
well. Figure 2A, please correct the borders are not uniform with Fig 2 B, C, D and A
denotation is not properly placed in the A figure.
5. Figure 4 the ‘genotype’ word is not visible, and some letters mixed with ‘BC182’. Please
put it appropriately in the figure and correct ‘S. tuberosum’ it is mixed with the mean weight
of tubers
6. Line no. 281 in the results section, ‘effect of drought stress on the photosynthesis’ it should be
subsection 3.5; please correct it
7. Figure 6 A and B, the figure is denoted by A and B at two places. Please remove one from the
graph, and in Figure B, S. bulbocastanum is changed with S.tubersosum and vice-versa in
Fig A, please make corrections in this figure
8. Line no. 312-313, where is the figure for the second measurement result of the efficiency of
water splitting complex (Fv/Fo), please add the figure and place it with fig 7 and give
denotation as fig 7A and B
9. Line no. 323, please correct it should be ‘figure 8A’

10. Figure 8A, in every figure, you have written S. bulbocastanum, but here it is Solanum
bulbocastanum, keep ‘S. bulbocastanum’ consistent in every figure and ‘S. tubersoum’ is
written in other figures, but here it is S. tubersoum CV Delikat; please correct this, and
keep S. tuberosum consistent in every figure
11. Rakosy-Tycan et al (2020) showed these somatic hybrids from S. tuberosum and S.
bulbocastanum is resistant to late blight infection, and in your study, it was proved that
these somatic hybrids demonstrate tolerance to drought stress, and you have not done any
studies to prove late blight infection resistance in these somatic hybrids, so I suggest to
make changes in the title, ‘remove resistant to late blight infection’ or modify in such a
the way that it should only reflect your results and study about drought stress tolerance
Major Comments: In this research, authors have concluded that these somatic hybrids are tolerant to drought
stress based on only a few parameters (largely physiological parameters), such as proline content,
biomass accumulation and drought stress effect on photosynthesis.  1. I think these parameters are not enough to prove drought-tolerant plants, so I would suggest the
authors to do check few more physiological parameters, such as the relative water content in
leaf, leaf electrolyte leakage, stomatal density, Antioxidant enzymatic activities in leaf
extracts, Malondialdehyde (MDA) content, etc. 2. Last but not least, if possible, some transcript studies related to drought stress.

Reviewer 3 Report

Comments and Suggestions for Authors

The study investigates drought tolerance and resistance to Late Blight Infection in somatic hybrids derived from common potato species and a wild potato species under in vitro conditions and PEG osmotic stresses. While the paper demonstrates scientific soundness and innovation, significant issues, especially in the materials and methods section and result presentation, render the article unacceptable without addressing these flaws.

The M&M section requires more extensive details, particularly about genotypes and plant materials. Essential specifications include whether somatic hybrids were prepared from a gene bank related to previous studies, the number of hybrids prepared, their parentage, their establishment in the culture medium (callus or seedlings), and information about resistance genes. Details can be presented in a tabular format.

-The methodology for evaluating Late Blight Infection needs explicit detailing.

-Biomass-related traits should be expressed as dry weight, regardless of the measurement method used. Clarification is needed in the biomass calculation section.

-Plant yield should ideally be quantified in weight per plant or per unit area for accurate performance assessment.

-Statistical analysis using t-tests for various analyses should be replaced with ANOVA analysis allowing multiple comparisons. Meaningful placement of statistical symbols in figures is advised for clarity. The conclusion section needs to be more comprehensive to match the study's depth and importance.

-The absence of the control group mentioned in the methods from the table affects data interpretation.

-Rewrite conclusion section with more details.

Reviewer 4 Report

Comments and Suggestions for Authors

The current title is misleading and could be replaced. 

There are numerous places needing corrections or additions. The comments are given in the included file.

Comments on the Quality of English Language

The English Language requires medium corrections.

Round 2

Reviewer 3 Report

Comments and Suggestions for Authors

The intended comments have been applied. In the title, the names of the authors are mixed with the title. figure 1 Y-axis title should be corrected; it mixed with axile numbers. figure 3 titles should be written in the section below.